# The Evaluation of Clot Waveform Analyses for Assessing Hypercoagulability in Patients Treated with Factor VIII Concentrate

**DOI:** 10.3390/jcm12196320

**Published:** 2023-09-30

**Authors:** Takeshi Matsumoto, Hideo Wada, Katsuya Shiraki, Kei Suzuki, Yoshiki Yamashita, Isao Tawara, Hideto Shimpo, Motomu Shimaoka

**Affiliations:** 1Department of Transfusion Medicine and Cell Therapy, Mie University Hospital, Tsu 514-8507, Japan; matsutak@clin.medic.mie-u.ac.jp; 2Department of General and Laboratory Medicine, Mie Prefectural General Medical Center, Yokkaichi 510-0885, Japan; katsuya-shiraki@mie-gmc.jp; 3The Emergency and Critical Care Center, Mie University Hospital, Tsu 514-8507, Japan; keis@med.mie-u.ac.jp; 4Department of Hematology and Oncology, Mie University Graduate School of Medicine, Tsu 514-8507, Japan; yamayamafan4989@yahoo.co.jp (Y.Y.); itawara@clin.medic.mie-u.ac.jp (I.T.); 5Mie Prefectural General Medical Center, Yokkaichi 510-0885, Japan; hideto-shimpo@mie-gmc.jp; 6Department of Molecular Pathobiology and Cell Adhesion Biology, Mie University Graduate School of Medicine, Tsu 514-8507, Japan; motomushimaoka@gmail.com

**Keywords:** FVIII, emicizumab, CWA, APTT, thrombin time, thrombin burst

## Abstract

Background: Regular prophylactic therapy has become an increasingly common treatment for severe hemophilia. Therefore, hypercoagulability—a potential risk factor of thrombosis—is a cause for concern in hemophilic patients treated with a high dose of FVIII concentrate. In clot waveform analysis (CWA)-thrombin time (TT), a small amount of thrombin activates clotting factor VIII (FVIII) instead of fibrinogen, resulting in FVIII measurements using CWA-TT with a small amount of thrombin. Methods: The coagulation ability of patients treated with FVIII concentrate or emicizumab was evaluated using activated partial thromboplastin time (APTT), TT and a small amount of tissue factor-induced FIX activation assay (sTF/FIXa) using CWA. Results: The FVIII activity based on CWA-TT was significantly greater than that based on the CWA-APTT or chromogenic assay. FVIII or FVIII-like activities based on the three assays in plasma without emicizumab were closely correlated; those in plasma with emicizumab based on CWA-TT and chromogenic assays were also closely correlated. CWA-APTT and CWA-TT showed different patterns in patients treated with FVIII concentrates compared to those treated with emicizumab. In particular, CWA-TT in patients treated with FVIII concentrate showed markedly higher peaks in platelet-rich plasma than in platelet-poor plasma. CWA-APTT showed lower coagulability in hemophilic patients treated with FVIII concentrate than in healthy volunteers, whereas CWA-sTF/FIXa did not. In contrast, CWA-TT showed hypercoagulability in hemophilic patients treated with FVIII concentrate. Conclusions: CWA-TT can be used to evaluate the thrombin bursts that cause hypercoagulability in patients treated with emicizumab. Although routine APTT evaluations demonstrated low coagulation ability in patients treated with FVIII concentrate, CWA-TT showed hypercoagulability in these patients, suggesting that the evaluation of coagulation ability may be useful when using multiple assays.

## 1. Introduction

Hemophilia is a congenital bleeding disorder characterized by decreased or defective clotting factor VIII (FVIII) or FIX levels [1,2]. Regular prophylactic treatment with FVIII concentrate, including extended half-life FVIII concentrate (EHL-FVIII), helps to prevent joint bleeding and damage in severe cases of hemophilia [1,2,3,4]. EHL-FVIII reduces the number of injections required by patients, as well as the amount of bleeding, substantially improving the treatment of severe hemophilia [4,5]. Recently, efanesoctocog alfa, a von Willebrand factor (VWF)-independent recombinant FVIII concentrate, was developed [6,7]. FVIII concentrates may elevate FVIII activity by more than 100–150% in hemophilic patients.

Emicizumab (Chugai Pharmaceutical Co., Ltd., Tokyo, Japan) [8,9], a bispecific antibody for FX and FIX, is useful for patients with life-threatening hemophilia, with or without inhibitors of FVIII, and can reduce the injection frequency for the treatment of hemophilia A. However, the measurement of FVIII activity and inhibitors is difficult to perform using the activated partial thromboplastin time (APTT) one-stage clotting assay in hemophilia A patients treated with emicizumab. Although hemophilic patients with inhibitors should be treated with bypass therapy at the time of major surgery or severe bleeding, FVIII inhibitors cannot be evaluated using the APTT one-stage clotting assay in patients treated with emicizumab. Therefore, anti-idiotype monoclonal antibodies for emicizumab have been established, and it has been deemed possible to measure FVIII activity and inhibitor titers using this antibody in the presence of emicizumab [10]. Furthermore, the new chromogenic substrate assay is not affected by emicizumab and is suitable for determining FVIII coagulant activity and inhibitors [11].

FVIII activity and coagulation ability have been analyzed in several reports using a routine APTT assay based on the peak times of clot waveform analysis (CWA)-APTT and chromogenic substrate assay [12,13,14]. However, few reports have described the relationship between FVIII activity, which is assessed using the peak time and height of CWA-APTT, including a small amount of tissue factor-induced activated FIX (sTF/FIXa) assay [15]. A small amount of thrombin reflects the thrombin burst, including the activation of FXI, FVIII, and FV instead of fibrinogen, and thrombin time (TT) using only a small amount of thrombin can evaluate the intrinsic pathway and FVIII activity [16]. Recently, it was reported that CWA-TT can measure FVIII activity independent of the presence of emicizumab [17].

In this study, we evaluated the coagulation ability and FVIII activity in 23 patients with hemophilia, 1 patient who was a carrier of hemophilia A, and 1 patient with acquired hemophilia A using CWA-APTT, CWA-TT, and chromogenic assays.

## 2. Materials and Methods

Twenty-eight plasma samples were obtained from twenty-three patients with hemophilia, one patient who was a carrier of hemophilia A, and one patient with acquired hemophilia A. The patients were managed at Mie University Hospital between 1 January 2022 and 31 December 2022 (Table 1). Twenty-four samples were obtained from patients treated with FVIII concentrate, emicizumab, or activated prothrombin complex concentrate (APCC). The interval between sample correction and treatment varied for each patient. Control plasma samples were obtained from 20 healthy volunteers (6 males and 14 females, aged 21–58 years). The study protocol (2249) was approved by the Human Ethics Review Committee of Mie University Hospital and signed informed consent was obtained from each participant. Twenty plasma samples from healthy volunteers were used as the normal reference plasma samples. This study was conducted in accordance with the principles of the Declaration of Helsinki.

The CWA-TT (Figure 1a) using 0.5 IU thrombin (Thrombin 500 units; Mochida Pharmaceutical Co., Ltd., Tokyo, Japan) was measured using an ACL-TOP^®^ system (Instrumentation Laboratory, Bedford, MA, USA) [16,17]. This system shows three types of curves [16,17]. One illustrates the changes in the absorbance observed while measuring CWA-TT, corresponding to the fibrin formation curve (FFC). The second is the first derivative peak (1st DP) of the absorbance, which corresponds to the coagulation velocity. The third is the second derivative peak (2nd DP) of the absorbance, which corresponds to the acceleration of coagulation. Calibration plasma (Instrumentation Laboratory) was used to study normal plasma. Emicizumab (Chugai Pharmaceutical Co., Ltd., Tokyo, Japan) was kindly provided by Chugai Pharmaceutical Co., Ltd.

CWA-APTT (Figure 1b) of platelet-poor plasma (PPP) was measured using a HemosIL APTT-SP (Instrumentation Laboratory), as previously reported [15]. PRP was prepared via centrifugation at 900 rpm for 15 min (platelet count, 40 × 10^10^ L), and PPP was prepared via centrifugation at 3000 rpm for 15 min (platelet count, <0.5 × 10^10^ L) [18].

The sTF/FIX assay (Figure 1c) was performed using PRP and 2000-fold diluted HemosIL RecombiPlasTin 2G (Instrumentation Laboratory) via the ACL-TOP^®^ system [19].

FVIII or FVIII-like activity was measured using the one-stage clotting assay of APTT peak time (normal reference range: median, 100% [25–75th percentile, 88.2–137%]) with APTT-SP, FVIII-deficient plasma (Instrumentation Laboratory), and the ACL-TOP system, with the chromogenic substrate method using a Revohem^TM^ FVIII chromogenic system (HYPHEN BioMed, Neuville-sur-Oise, France) using a CS-5100 device (Sysmex Corporation, Kobe, Japan), or with the CWA-TT method (normal reference range: 104% [88.2–137%]) using FVIII-deficient plasma (Instrumentation Laboratory) and an ACL-TOP system [17].

### Statistical Analyses

Data are expressed as the median (25–75th percentiles). The significance of the differences between the groups was examined using the Mann–Whitney *U*-test. *p* < 0.05. All statistical analyses were performed using the Stat-Flex software (version 6; Artec Co., Ltd., Osaka, Japan).

## 3. Results

Of the 23 patients with hemophilia, 15 were treated with regular replacement therapy, 7 with EHL-FVIII, and 5 with emicizumab (Table 1). The two standard curves (Y = 0.284X − 94.3 vs. Y = 0.291X − 96.7) for FVIII activity in normal reference plasma from 20 healthy volunteers using the CWA-TT in plasma with and without emicizumab (final concentration, 0.15 mg/mL) were similar (difference ≤ 1%) (Figure 2), suggesting that a standard curve in plasma without emici, significantly zumab, could be used to determine FVIII activity in plasma with emicizumab. In contrast, the two standard curves (Y = 225 − 1.78X vs. Y = 171 − 3.91X) for FVIII activity using the APTT one-stage clotting assay in plasma with and without emicizumab differed. FVIII activities in PPP from patients treated without emicizumab based on the one-stage clotting assay using the CWA-TT were significantly higher than those based on the one-stage clotting assay using a CWA-APTT and a chromogenic assay (Figure 3). Although the FVIII activity using the APTT one-stage clotting assay was scaled over in plasm with emicizumab, the activities based on CWA-APTT and CWA-TT in plasma without emicizumab were closely correlated, and those based on CWA-TT and a chromogenic assay in plasma with and without emicizumab were also closely correlated (Figure 4a–c).

CWA-APTT in HA-2 treated with FVIII concentrate (efraloctocog alfa; Sanofi K.K., Tokyo, Japan) showed that the peak time was prolonged, and the peak height was similar to that of the normal control (Figure 5a), whereas CWA-APTT in HA-22 treated with emicizumab showed shorter peak time and relatively lower peak height compared with the normal control (Figure 5b). HA-3 treated with FVIII concentrate (rurioctocog alfa pegol; Takeda Pharmaceuticals, Osaka, Japan) showed that FVIII activities reached 7.4–12.4%, with a prolonged peak time and relatively low peak height on CWA-APTT and shortened peak time and normal peak height of 1st DP on CWA-TT (Figure 6a). HA-2 treated with FVIII concentrate (efraloctocog alfa) showed FVIII activity of 16.3–21.0%, with a slightly prolonged peak time and normal peak height on CWA-APTT and a shortened peak time and elevated peak height of 1st DP on CWA-sTF/FIXa (Figure 6b). HA-10 treated with FVIII concentrate (Lonoctocog alfa; CSL Behring K.K., Tokyo, Japan) showed that the 1st DPH and 2nd DPH were similar to the values in healthy volunteers in the CWA-APTT, and the 1st DPT and 2nd DPT were shortened, while the 1st DPH was similar to that in healthy volunteers in the CWA-sTF/FIXa, indicating that the correlation of FVIII activity with the CWA differed.

Regarding CWA-TT, HA-6 treated with emicizumab showed a low peak height of the 1st DP and no marked difference between PPP and PRP (Figure 7a,b), whereas HA-2 treated with FVIII concentrate showed a low peak height and second peak of the 1st DP in PPP, and a markedly high peak height and combined first and second peaks of the 1st DP in PRP (Figure 7c,d).

The peak times (2nd DP [hemophilic patients treated with FVIII concentrate vs. healthy volunteers: 42.4 s; 38.4–51.2 s vs. 31.7 s; 31.1–32.9 s; *p* < 0.001], 1st DP [58.8 s; 43.7–59.1 s vs. 34.3 s; 33.8–35.9 s; *p* < 0.001] and FFC [53.2 s; 44.8–60.1 s vs. 36.0; 35.6–38.2 s; *p* < 0.001]) of CWA-APTT were significantly longer in hemophilic patients treated with FVIII concentrate than in healthy volunteers; however, peak times of CWA-sTF/FIXa (2nd DP [45.9 s; 38.4–51.2 s vs. 76.9 s; 70.6–84.0 s; *p* < 0.001], 1st DP [69.9 s; 62.8–75.8 s vs. 95.3 s; 89.8–106 s; *p* < 0.001] and FFC [71.9 s; 65.4–77.8 s vs. 94.6; 89.4–109 s; *p* < 0.001]) and CWA-TT (2nd DP [29.3 s; 28.0–38.1 s vs. 43.0 s; 40.8–46.5 s; *p* < 0.001], 1st DP [32.9 s; 31.7–43.0 s vs. 50.1 s; 48.2–54.3 s; *p* < 0.001] and FFC [51.8 s; 47.7–70.6 s vs. 136; 138–147 s; *p* < 0.001]) were significantly shorter in hemophilic patients treated with FVIII concentrate than in healthy volunteers (Figure 8). The peak heights of 2nd DP and 1st DP on the CWA-APTT were significantly lower in hemophilic patients treated with FVIII concentrate in comparison to healthy volunteers; however, those of FFC in the CWA-APTT (hemophilic patients treated with FVIII concentrate vs. healthy volunteers: 266 mm absorbance; 216–326 mm absorbance vs. 187 mm absorbance; 170–206 mm absorbance; *p* < 0.001), 2nd DP in the CWA-sTF/FIXa (78.1 mm absorbance; 59.9–93.9 mm absorbance vs. 37.4 mm absorbance; 31.0–48.1 mm absorbance; *p* < 0.001), 2nd DP (451 mm absorbance; 329–545 mm absorbance vs. 103 mm absorbance; 86.0–113 mm absorbance; *p* < 0.001), 1st DP (222 mm absorbance; 161–257 mm absorbance vs. 82.6 mm absorbance; 75.3–94.9 mm absorbance, *p* < 0.001), and FFC (791 mm absorbance; 687–991 mm absorbance vs. 696 mm absorbance; 622–727 mm absorbance; *p* < 0.001) in the CWA-TT were significantly higher in hemophilic patients treated with FVIII concentrate in comparison to healthy volunteers. There were no significant differences in the peak heights of 1st DP and FFC on CWA-sTF/FIXa between hemophilic patients and healthy volunteers (Figure 9).

## 4. Discussion

Regular prophylactic therapy including EHL-FVIII and emicizumab is common [20,21]. Treatment with emicizumab has become increasingly widespread [22,23], as emicizumab is quick and easy to inject. However, FVIII activity and inhibitors cannot be measured using routine APTT assays in patients treated with emicizumab [24]. Preincubation of plasma with APTT reagent and emicizumab before the addition of calcium solution forms FIXa, emicizumab, and FX complexes, resulting in shortened APTT. Therefore, we developed an FVIII assay based on CWA-TT, which reflects thrombin burst, independent of the presence of emicizumab [17]. Although the correlation coefficients of the three FVIII activities in plasma without emicizumab based on the CWA-APTT, CWA-TT or chromogenic assay were high, FVIII activities based on CWA-TT were two-fold higher than those based on a CWA-APTT or chromogenic assay, suggesting that the FVIII concentration may be easily affected by thrombin burst [18,25], exhibiting increased FVIII activity based on CWA-TT. EHL-FVIII can also decrease the number of injections required; however, FVIII activity may exceed 100% in hemophilic patients treated with EHL-FVIII. In particular, efanesoctocog alfa can maintain 5% FVIII activity for seven days after administration, with a maximum FVIII concentration of 150% [6]. Elevated FVIII activity was reported as a risk factor for venous thromboembolism [26,27]. Several reports have described thrombotic risk in hemophilic patients [28,29]. Although a high dose of FVIII-EHL can increase FVIII activity in hemophilic patients, as well as in normal individuals, injection with a high dose of FVIII-EHL is a regularly recent treatment method. Hemophilic patients treated with high-dose FVIII-EHL are at low risk of thrombosis in the short term; however, these patients may have a thrombotic risk close to that of healthy individuals in the long run.

FVIII activity was assessed via the APTT one-stage clotting assay [30]. However, this method is not the gold standard for measuring the FVIII activity. FVIII activity based on CWA-TT may be enhanced by a thrombin burst from 150% to ≥200% in patients treated with efanesoctocog alfa [31,32]. CWA-APTT in patients treated with emicizumab showed a shortened peak time; however, this shortness of CWA-APTT does not reflect physiological coagulation ability. Although combination therapy with emicizumab and APCC has been reported to be associated with thrombosis [8,33], monotherapy with emicizumab rarely leads to thrombotic complications. The peak height on the CWA-APTT may reflect physiological coagulation ability, and a low peak height on the CWA-APTT has been associated with major bleeding [15]. In addition, the peak height of CWA-APTT may reflect FVIII activity [34,35]. Elevated peak heights in CWA-APTT and CWA-sTF/FIXa have been associated with thrombosis [15], suggesting that greater peak heights in CWA-APTT and CWA-sTF/FIXa may indicate a risk of thrombosis. This increase in peak height for CWA-APTT and CWA-sTF/FIXa may be due to thrombin burst [18,36]. These findings suggest that the peak height on a CWA may be a more useful parameter than the peak time on a CWA, such as routinely measured APTT.

Regarding hemophilic patients treated with FVIII concentrate, the number of patients included in this study was relatively small; however, these CWA analyses showed significant findings regarding coagulability in these patients. CWA-APTT showed hypocoagulability in hemophilic patients treated with FVIII concentrate; however, CWA-sTF/FIXa did not. In contrast, CWA-TT exhibited hypercoagulability in hemophilic patients treated with FVIII concentrates. These differences are the result of preincubation with APTT regent, which activates FXII and, subsequently, FIX (FIXa), disturbing the evaluation of the activation ability of FXII to FIXa and reflection of thrombin burst (Figure 10). In CWA-sTF/FIX, a small amount of the TF and FVIIa complex activates FIX to FIXa, which reflects the thrombin burst. In CWA-TT, a small amount of thrombin activates FXI, FVIII, and FV instead of fibrinogen, which is also reflected by the thrombin burst. These findings suggest that some hemophilic patients treated with FVIII concentrate, who are evaluated for hypocoagulability using the routine APTT assay, may be at increased risk of thrombosis, as evaluated based on CWA-TT. In addition, the peak height of CWA-TT was significantly higher in PRP than in PPP, suggesting that hemophilic patients treated with FVIII concentrate, especially EHL-FVIII concentrate, were affected by thrombin burst-dependent platelets. These findings suggest that hemophilic patients treated with FVIII concentrate may temporarily be at the same or a higher risk of thrombosis than healthy individuals. This raises the question of whether the temporal effect of high-dose FVIII concentrate is problematic for thrombotic risk in hemophilic patients.

Limitation and importance: The present study was retrospective and small. In addition, FVIII concentrates and the intervals between the administration and blood sampling times varied. However, despite the small size and heterogenous nature of our study, the results showed statistical significance, suggesting that the findings are significant. Hemophilic patients treated with FVIII concentrate showed hypocoagulability, according to the prolonged peak time of CWA-APTT, but might also show hypercoagulability according to the shortened peak time and an elevated peak height of CWA-TT.

## 5. Conclusions

The FVIII activity in the present study was evaluated using several methods, including APTT, CWA-TT, and chromogenic assays, and the APTT one-stage method is not the gold standard. All parameters of CWA-TT and several parameters of CWA-sTF/FIXa showed that hypercoagulability was affected by thrombin bursts in hemophilic patients treated with FVIII concentrate. This raises the question of whether a transient increase in FVIII activity leads to a thrombotic risk.

## Figures and Tables

**Figure 1 jcm-12-06320-f001:**
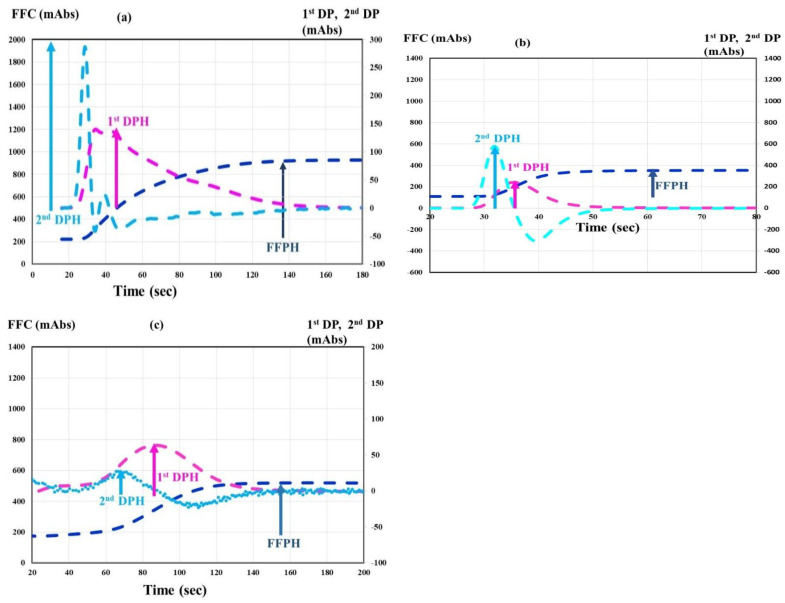
CWA-TT (**a**), CWA-APTT (**b**), and CWA-sTF/FIXa, (**c**) in a healthy volunteer. CWA—clot waveform analysis; APTT—activated partial thromboplastin time; navy line—fibrin formation curve; FFC—fibrin formation curve; pink line—1stDP—1st derivative peak (velocity); light blue—2nd DP 2nd derivative peak (acceleration); FFPH—peak height of FFC; 1st DPH—peak height of 1st DP; 2nd DPH—peak height of 2nd DP.

**Figure 2 jcm-12-06320-f002:**
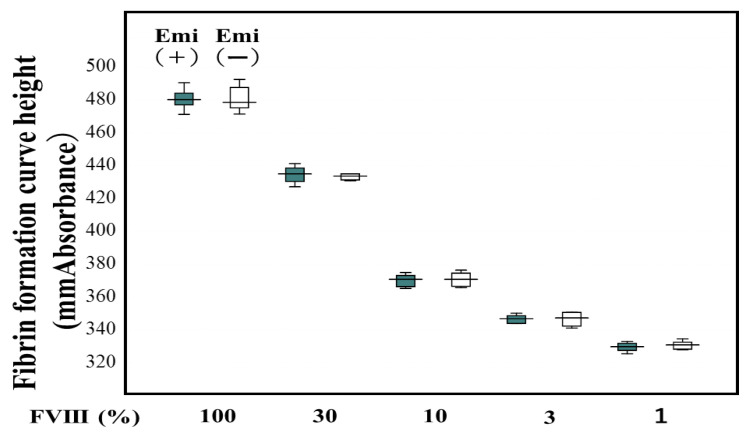
Standard curve for the FVIII and FVIII-like activity in normal reference plasma from healthy volunteers with and without emicizumab. Emi—emicizumab; FVIII—coagulation factor FVIII; closed box—FVIII activity with emicizumab (final concentration—0.15 mg/mL); open box—FVIII activity without emicizumab. Standard curves (Emi(+), Y = 0.284X − 94.3; Emi (−), Y = 0.291X − 96.7).

**Figure 3 jcm-12-06320-f003:**
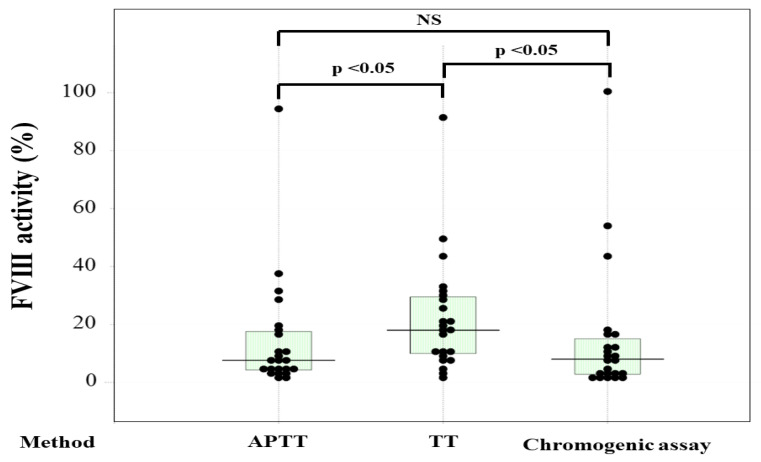
The FVIII activity based on a one-stage clotting assay using the clot waveform analysis (CWA)-APTT or CWA-thrombin time or a chromogenic assay. FVIII—coagulation factor FVIII; APTT—activated partial thromboplastin time; TT—thrombin time; FVIII activity without emicizumab; NS—not significant. Plasma with emicizumab was excluded for measurement of FVIII activity.

**Figure 4 jcm-12-06320-f004:**
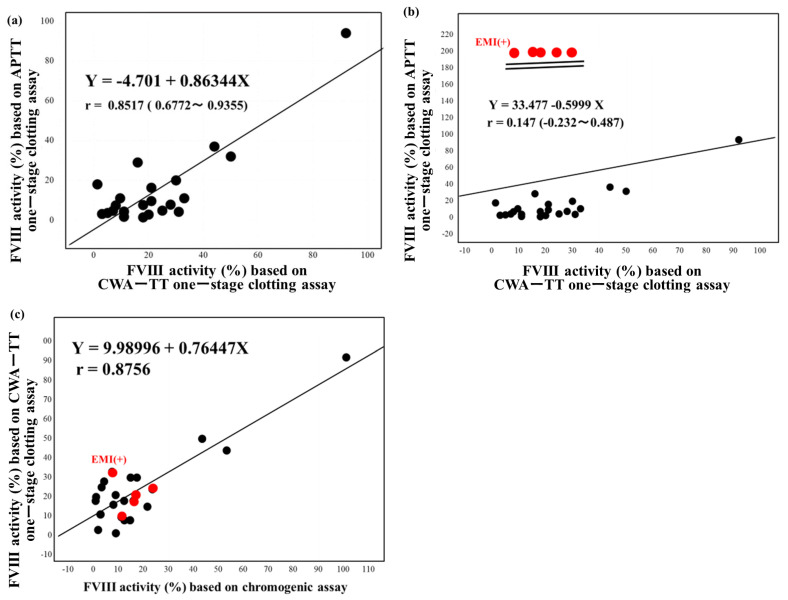
Correlation between the FVIII and FVIII-like activity based on APTT one―stage clotting assay using the APTT and CWA―thrombin time (**a**), without emicizumab and (**b**), with emicizuma (**b**) and between the FVIII activity based on a one―stage clotting assay using the CWA―thrombin time and a chromogenic assay (**c**). FVIII—coagulation factor FVIII; CWA—clot waveform analysis; APTT—activated partial thromboplastin time; TT—thrombin time; Emi—emicizumab; red symbol—plasma with emicizumab.

**Figure 5 jcm-12-06320-f005:**
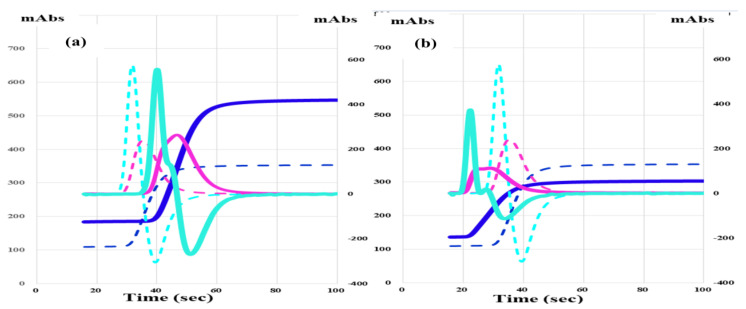
CWA―APTT in a hemophilia patient treated with FVIII concentrate (**a**) and a hemophilia patient treated with emicizumab (**b**). CWA—clot waveform analysis; APTT—activated partial thromboplastin time; navy line—fibrin formation curve; FFH—fibrin formation height; pink line—1st derivative curve (velocity); 1st DPH—first derivative peak height; light blue—2nd derivative curve (acceleration); 2nd DPH—second derivative peak height; solid line—patient; dotted line—healthy volunteer. Difference between the FVIII concentrate (**a**) and emicizumab (**b**); the peak times were prolonged, but the peak heights were within normal limits in (**a**), whereas the peak times were shortened but peak heights were low in (**b**).

**Figure 6 jcm-12-06320-f006:**
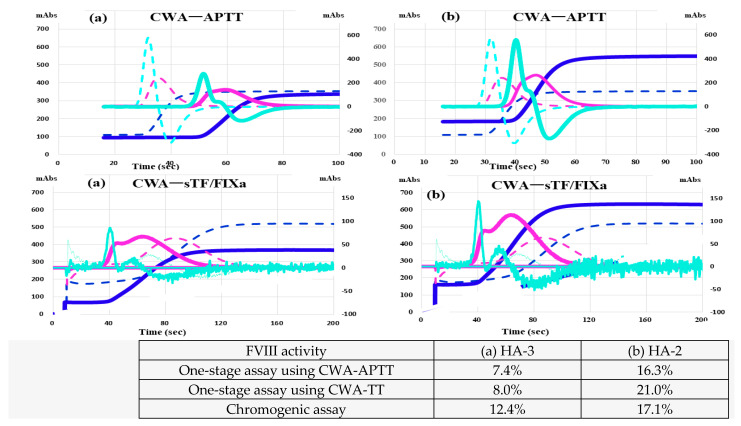
CWA―APTT and CWA―sTF/FIXa in a hemophilia patient treated with FVIII concentrate. CWA—clot waveform analysis; APTT—activated partial thromboplastin time; sTF/FIXa—small amount of tissue factor-induced FIX activation assay; navy line—fibrin formation curve; FFH—fibrin formation height; pink line—1st DP, 1st derivative (velocity); 1st DPH, 1st DP height; 1st DPT time, light blue—2nd derivative peak—2nd DP (acceleration); 2nd DPH—2nd DP height; 2nd DPT—2nd DP time; solid line—patient; dotted line—healthy volunteer. (**a**) The FVIII activity was low, but the 1st DPT and 2nd DPT were shortened, and 1st DPH was similar to that in healthy volunteers in the CWA-sTF/FIXa. (**b**) The FVIII activity was low, but the 1st DPH and 2nd DPH were similar to those in healthy volunteers in the CWA-APTT, and the 1st DPT and 2nd DPT were shortened, while the 1st DPH was similar to that in healthy volunteers in the CWA-sTF/FIXa. The correlation between FVIII activity and CWA-APTT and CWA-sTF/FIX differed.

**Figure 7 jcm-12-06320-f007:**
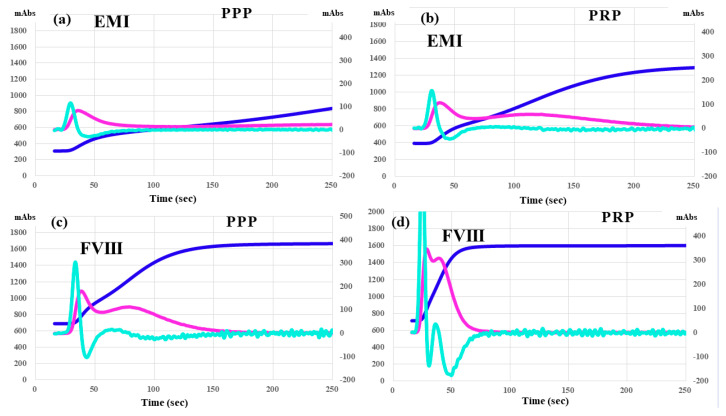
CWA—TT in a hemophilia patient treated with emicizumab (**a**,**b**) or FVIII concentrate (**c**,**d**). CWA—clot waveform analysis; TT—thrombin time; navy line—fibrin formation curve; FFH—fibrin formation height; pink line—1st derivative curve (velocity); 1st DPH—first derivative peak height; light blue—2nd derivative curve (acceleration); 2nd DPH—second derivative peak height; PPP—platelet poor plasma (**a**,**c**); PRP—platelet-rich plasma (**b**,**d**); EMI. Emicizumab; FVIII—coagulation factor FVIII reagent.

**Figure 8 jcm-12-06320-f008:**
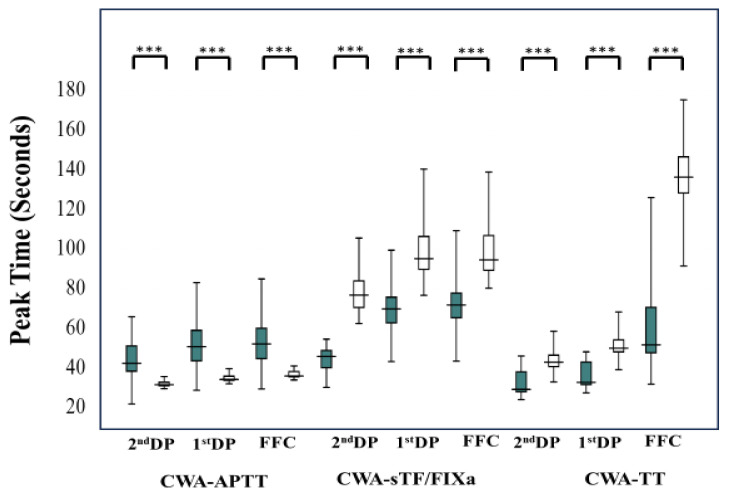
Peak times of CWA-APTT, CWA-sTF/FIXa, and CWA-TT in hemophilia patients treated with FVIII concentrate. CWA—clot waveform analysis; APTT—activated partial thromboplastin time; sTF/FIXa—small amount of tissue factor-induced FIX activation assay; TT—thrombin time; FFC—fibrin formation curve; 1st DP—1st derivative peak; 2nd DP—second derivative peak; ***—*p* < 0.001; closed box—hemophilia patients treated with FVIII concentrate; open box—healthy volunteers.

**Figure 9 jcm-12-06320-f009:**
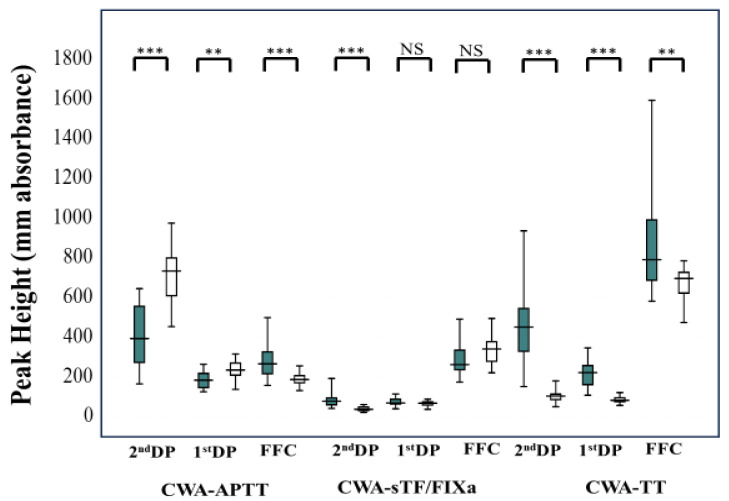
Peak heights of CWA-APTT, CWA-sTF/FIXa, and CWA-TT in hemophilia patients treated with FVIII concentrate. CWA—clot waveform analysis; APTT—activated partial thromboplastin time; sTF/FIXa—small amount of tissue factor-induced FIX activation assay; TT—thrombin time; FFC—fibrin formation curve; 1st DP—1st derivative peak; 2nd DP—second derivative peak; ***—*p* < 0.001; **—*p* < 0.01; NS—not significant; closed box—hemophilia patients treated with FVIII concentrate; open box—healthy volunteers.

**Figure 10 jcm-12-06320-f010:**
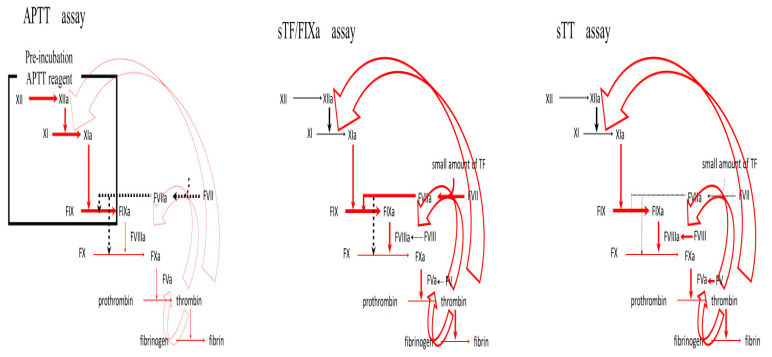
Differences among APTT, sTF/FIXa, and sTT assays. APTT—activated partial thromboplastin time; sTF/FIXa—small amount of tissue factor-induced FIX activation assay; sTT—small amount of thrombin time; red arrow—coagulation process; open red arrow—thrombin burst.

**Table 1 jcm-12-06320-t001:** Subjects.

No	Disease	Severity	Basal FVIII Activity	Inhibitor	RRT	Drug
HA-1	HA	Severe	≤1.0%	Negative	Yes	Rurioctocog alfa pegol
HA-2	HA	Moderate	1.8%	Negative	Yes	Efraloctocog alfa
HA-3	HA	Severe	≤1.0%	Negative	Yes	Rurioctocog alfa pegol
HA-4	HA	Severe	≤1.0%	Negative	Yes	Rurioctocog alfa pegol
HA-5	HA	Mild	23.3%	Negative	No	Octocog beta
HA-6	HA	Severe	≤1.0%	Positive	Yes	Emicizumab
HA-7	HA	Severe	≤1.0%	Negative	Yes	Octocog beta
HA-8	HA	Moderate	≤1.0%	Negative	No	Rurioctocog alfa pegol
HA-9	HA	Severe	≤1.0%	Negative	Yes	Emicizumab
HA-10	HA	Severe	≤1.0%	Negative	Yes	Lonoctocog alfa
HA-11	HA	Severe	1.0%	Negative	Yes	Octocog beta
HA-12	HA	Mild	14.1%	Negative	No	FDCHB-FVIII
HA-13	HA	Severe	≤1.0%	Negative	Yes	Rurioctocog alfa pegol
HA-14	HA	Moderate	1.6%	Negative	No	―
HA-15	HA	Moderate	3.9%	Negative	No	Rurioctocog alfa pegol
HA-16	HA	Mild	6.7%	Negative	No	Rurioctocog alfa
HA-17	HA	Mild	5.4%	Negative	No	Rurioctocog alfa
HA-18	HA	Moderate	2.2%	Negative	No	―
HA-19	HA	Severe	≤1.0%	Negative	Yes	Rurioctocog alfa pegol
HA-20	HA	Severe	≤1.0%	Negative	Yes	Emicizumab
HA-21	HA	Moderate	1.0%	Negative	Yes	Emicizumab
HA-22	HA	Severe	≤1.0%	Negative	Yes	Emicizumab
HA-23	HA	Severe	≤1.0%	Negative	Yes	Efraloctocog alfa
24	HA *	Mild	20.4%	Negative	No	―
25-1	AHA	Severe	≤1.0%	Positive	No	APCC
25-2	AHA	Severe	≤1.0%	Positive	No	APCC
25-3	AHA	Severe	≤1.0%	Positive	No	APCC
25-4	AHA	Severe	≤1.0%	Positive	No	―

HA *—HA carrier; RRT—regular replacement therapy; HA—hemophilia; FVIII—coagulation factor FVIII; AHA—acquired HA; APCC—activated prothrombin complex concentrate; FDCHB-FVIII—freeze-dried concentrated human blood-FVIII; Basal FVIII activity was measured in patients who were not treated with FVIII concentrate.

## Data Availability

The data presented in this study are available upon request from the corresponding author. The data are not publicly available due to privacy restrictions.

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
