# Peer review of "The Evaluation of Clot Waveform Analyses for Assessing Hypercoagulability in Patients Treated with Factor VIII Concentrate"

_jcm, 2023, doi:10.3390/jcm12196320_

Round 1

Reviewer 1 Report (Previous Reviewer 3)

The manuscript has been corrected, however I do not have the insight to reviewers' comments, and cannot conclude if all comments have been addressed.

The manuscript read better now, more details have been included.

However, several points need to be addressed before publishing:

The sentence in the abstract needs correction: "A samll amount of thrombin activates clotting factor VIII (FVIII) in stead of fibrinogen, resulting in the development of FVIII meaurements using a clot waveform analysis (CWA) and a thrombin time (TT) uing small amount of thrombin." Besides typos, the massage is not clear.

"CWA-TT can be used to evaluate coagulability 34 in patients treated with emicizumab." I do not see the basis for this conclusion.

"FVIII and FVIII-like activities based on the three assays in plasma..." which tests are those three?

The authors are stating that "...plasma with emicizumab based on CWA-TT and chromogenic assays were also closely correlated." I do not understand why stating "also" if they describe two sentences before that FVIII activity based on TT is much higher than one measured by chromogenic assay?

In discussion, this sentence needs to be corrected: "The FVIII activity based on CWA-TT may be 200% in 248 patients treated with efanesoctocog alfa." Patients treated with factor concentrate (of any kind) may achieve any FVIII activity depending on the dosing of the factor concentrate.

The authors did a substantial changes in the manuscript. I cannot tell if all the comments have been addressed - I could not access the previous reviewers comments.

I would suggest adding additional minor changes before publishing.

Author Response

Response to Reviewer-1

The manuscript has been corrected, however I do not have the insight to reviewers' comments, and cannot conclude if all comments have been addressed.

The manuscript read better now, more details have been included.

However, several points need to be addressed before publishing:

Comment 1.                                                          The sentence in the abstract needs correction: "A samll amount of thrombin activates clotting factor VIII (FVIII) in stead of fibrinogen, resulting in the development of FVIII meaurements using a clot waveform analysis (CWA) and a thrombin time (TT) uing small amount of thrombin." Besides typos, the massage is not clear.

Response 1. This sentence has been revised as follows: “In the clot waveform analysis (CWA)-thrombin time (TT), a small amount of thrombin activates clotting factor VIII (FVIII) instead of fibrinogen, resulting in obtaining FVIII measurements using a CWA-TT with a small amount of thrombin.”

Comment 2.                                                       "CWA-TT can be used to evaluate coagulability 34 in patients treated with emicizumab." I do not see the basis for this conclusion.

Response 2. This sentence has been revised as follows: “The CWA-TT can be used to evaluate thrombin burst causing hypercoagulability in patients treated with emicizumab”.

Comment 3.                                                        "FVIII and FVIII-like activities based on the three assays in plasma..." which tests are those three?

Response 3. “FVIII and FVIII-like” has been revised as follows: FVIII or FVIII-like”. The CWA-APTT, CWA-TT and chromogenic assay are the three assays.

Comment 4.

The authors are stating that "...plasma with emicizumab based on CWA-TT and chromogenic assays were also closely correlated." I do not understand why stating "also" if they describe two sentences before that FVIII activity based on TT is much higher than one measured by chromogenic assay?

Response 4. This sentence has been revised as follows: “the correlation coefficients of the three FVIII activities in plasma without emicizumab based on the CWA-APTT, CWA-TT or chromogenic assay were high”.

Comment 5.                                                           In discussion, this sentence needs to be corrected: "The FVIII activity based on CWA-TT may be 200% in 248 patients treated with efanesoctocog alfa." Patients treated with factor concentrate (of any kind) may achieve any FVIII activity depending on the dosing of the factor concentrate.

Response 5. This sentence has been revised.

Comment 6.

Comments on the Quality of English Language

The authors did a substantial changes in the manuscript. I cannot tell if all the comments have been addressed - I could not access the previous reviewers comments.

I would suggest adding additional minor changes before publishing.

Response 6. We have fully revised this manuscript.

Reviewer 2 Report (New Reviewer)

This article by Matsumoto et al describes a potential new tool to measure coagulability in haemophilia A patients treated with factor VIII or emicizumab. The new treatment modalities for haemophilia entering the market improve the live of our patients substantially, but also poses us a treaters for a new challenge of how to adequately monitor treatment especially in case of bleeding or surgery.

In their manuscript plasma samples from different patient groups are evaluated: congenital haemophilia A patients with (just one) and without inhibitors and different treatments, acquired haemophilia A patients. Also the interval wbetween sample correction and treatment varied for each patient. As such it is a very heterogeneic study and I feel interpreting the data is difficult. The description in the result section does not help me a lot. 

The result section starts with describing the results of plasma samples from healthy volunteers with and without emicizumab, something that is not described in the method section. Thereafter, the FVIII activity measured with different methods is shown in figure 3, and although showing differences between the used methods, which is important to recognize, none of these acitivities seem concerning to me.

Then, I get lost. Results of individual patients are described. It is unclear to me why this is described on an individual level and not on the above mentioned group level. I would be most concerned about thrombosis in acquired haemophilia A patients, but I cannot extract the data for these patients clearly.

Also the discussion is not written well. Thrombosis is not a serious problem in haemophilia patients on prophylaxis, the reference [22] describes the FVIII activity in VTE in non-haemophilia patients. It might be a problem in emicizumab + FEIBA treated patients, or in efanesoctocog alfa patients, but the second group is not included in this study.

The manuscript contains a lot of typo's or incorrect language, f.i. 

abstract line 3: samll

Introduction page 2 line 53: 'life threatining hemophilic patients'??

Page 2 line 71:  a small amount of thrombin time ??

Page 5 line 140 pateitns

Page 10 line 242: 'decreased' should be 'decrease'

Author Response

Response to Reviewer 2

Comments and Suggestions for Authors

This article by Matsumoto et al describes a potential new tool to measure coagulability in haemophilia A patients treated with factor VIII or emicizumab. The new treatment modalities for haemophilia entering the market improve the live of our patients substantially, but also poses us a treaters for a new challenge of how to adequately monitor treatment especially in case of bleeding or surgery.

Comment 1.                                                           In their manuscript plasma samples from different patient groups are evaluated: congenital haemophilia A patients with (just one) and without inhibitors and different treatments, acquired haemophilia A patients. Also the interval wbetween sample correction and treatment varied for each patient. As such it is a very heterogeneic study and I feel interpreting the data is difficult. The description in the result section does not help me a lot.

Response 1. The heterogenicity of patients treated with FVIII concentrate has been mentioned as a limitation. However, while the population of congenial hemophilic patients treated with FVIII concentrate in our study was a heterogenic group, we obtained statistically significant findings using a CWA-TT, suggesting that these results are strong and important.

Comment 2.                                                          The result section starts with describing the results of plasma samples from healthy volunteers with and without emicizumab, something that is not described in the method section. Thereafter, the FVIII activity measured with different methods is shown in figure 3, and although showing differences between the used methods, which is important to recognize, none of these acitivities seem concerning to me.

Response 2. The Material and Methods section has been revised. An explanation for the three assays has already been included in the Discussion with Figure 10.

Comment 3.                                                        Then, I get lost. Results of individual patients are described. It is unclear to me why this is described on an individual level and not on the above mentioned group level. I would be most concerned about thrombosis in acquired haemophilia A patients, but I cannot extract the data for these patients clearly.

Response 3. We already showed hypercoagulability in this group (congenital hemophilia patients treated with FVIII concentrate) in Figure 8 and 9 and also mentioned it in the Abstract and Discussion section. Regarding to thrombosis, TMA and DVT may cause its acute onset in cases of acquired hemophilia; however, the onset of cerebral infarction, myocardial infarction and chronic DVT may take a long time in hemophilic patients treated with FVIII concentrate. However, the history of injection with a high-dose of FVIII-EHL have only been prescribed relatively recently. We have now touched on this in the Discussion.

Comment 4.                                                         Also the discussion is not written well. Thrombosis is not a serious problem in haemophilia patients on prophylaxis, the reference [22] describes the FVIII activity in VTE in non-haemophilia patients. It might be a problem in emicizumab + FEIBA treated patients, or in efanesoctocog alfa patients, but the second group is not included in this study.

Response 4. The following paragraph has been added to the Discussion. A few     reports have described thrombotic risk in hemophilic patients [27, 28] Although a high dose of FVIII-EHL can increase FVIII activity as well as in normal individual, the history of injection with a high-dose of FVIII-EHL have only been prescribed relatively recently. Hemophilic patients treated with high-does FVIII-EHL have a low risk of thrombosis in the short term; however, these patients may have a thrombotic risk close to that of healthy individuals in the long term.

Comments on the Quality of English Language

The manuscript contains a lot of typo's or incorrect language, f.i.

Comment. abstract line 3: small  

Response. This word was revised to “small”.

Comment. Introduction page 2 line 53: 'life threatining hemophilic patients'??

Response. This sentence has been revised as follows: “life-threatening bleeding in hemophilic patients---”.

Comment. Page 2 line 71: a small amount of thrombin time ??

Response. This sentence has been revised as follows: “thrombin time (TT) using a small amount of thrombin can evaluate the intrinsic pathway”

Comment. Page 5 line 140 pateitns

Response. The word has been revised to “patients”.

Comment. Page 10 line 242: 'decreased' should be 'decrease'

Response. “Decreased” has been revised to “decrease”.

Round 2

Reviewer 2 Report (New Reviewer)

Apart from improving the English language no additional comments

There are too many English flaws in this article, I mentioned some in the previous review, but it is not my task to outline them all. 

By reading title and abstract only already three identified;

Title - 'for the assessing hypercoagulability' should be 'for assessing the hypercoagulability'

Abstract -Background - first sentence is unreadible

Abstract - Conclusions 'to evaluate thrombin bursts that causing ...' should be 'that cause...'

Please carefully read the entire article and improve the English language.

Author Response

There are too many English flaws in this article, I mentioned some in the previous review, but it is not my task to outline them all. 

By reading title and abstract only already three identified;

Comment 1.

Title - 'for the assessing hypercoagulability' should be 'for assessing the hypercoagulability'

Response 1. Title has been revised.

Comment 2.

Abstract -Background - first sentence is unreadible

Response 2. Background has been revised.

Comment 3.

Abstract - Conclusions 'to evaluate thrombin bursts that causing ...' should be 'that cause...'

Response 3. Conclusions has been revised.

Comment 4.

Please carefully read the entire article and improve the English language.

Response 4. Revised manuscript has been checked by MDPI English editing service.

This manuscript is a resubmission of an earlier submission. The following is a list of the peer review reports and author responses from that submission.

Round 1

Reviewer 1 Report

no further comment. the authors have adequately responded to previous critiques. 

Author Response

Comment

no further comment. the authors have adequately responded to previous critiques. 

Response: Thank you for your comment.

Reviewer 2 Report

The authors describe their work using clot wave analysis in Hemophilia A patients and various therapies. I found this manuscript difficult to follow the intent and purpose.  There are bullets of information provide, but I have to try and connect those dots.  

The claim that there is no mechanism for measuring inhibitors in patients on emicizumab is false, as chromogenic method using alternative factor has been described.

Unclear whether the factor VIII levels present in the table are upon diagnosis or measured during the study.  If the former, that information is not germance.

Unclear when the patients were treated with the therapy in relationship to the drug level, as many FVIII levels were really low, so it appears that these are random collections, versus peak or trough collections.

Materials and methods are poorly organized.  Describe the sample collection first, the sample processing next.  Indicate when the samples were tested in relationship to collection time.

If surrogate or contrive samples were created, including calibration curves, describe how that was done.  Unclear what emicizumab concentrations were assessed in this study, and how they were assessed.  How much emicizumab was in the calibration curve (Figure 1)

Next describe the testing methods.  As in reading throughout this document, it appears on the peak height is germane to this manuscript.  As such, provide a figure and indicate with an arrow, which peak height is being used for this purpose.

The statistical analysis is either not well described, or was not provided.  There are numerous indications where "significant" was used, yet no numeric or statistical information provided to support the use of that statistical term.

Unclear what defines "almost similar" regarding calibration curve described in Figure 1.  It is unclear how this was generated.  Was the same calibrator material used?  IF so, why does the x-axis reflect differences in calibrator value?  If not, then why not?

Figure 2 appears to demonstrate equivalence between acceptable FVIII measurements (chromogenic) and the proposed platforms.  However, correlation, slope are more useful (and required measures by regulatory authorities) that this simple plot.

Unclear the purpose for Figure 3a or 3b.  

Unclear all the lines around the correlation line.

As it appears these methods for ascertaining FVIII are different, what is the normal reference range for these platforms when using the same donors?

Figure 4: which are of the CWA is used for FVIII analysis.  What is the purpose of this curve other to demonstrate different CWA between the two patients?  Why list all the other clot waves if they are not part of the analysis?   Easier to distinguish clot waves if more than different shades of blue and red are used. 

Figure 5. The bottom curves need more explanation including where did this data come from as the navy (blue) line makes no sense as presented.  Is this a single HA patient with different FVIII over time after therapy?  Why is the overlay "a healthy volunteer" and not reflective of a normal range?  What is the purpose of this presentation?

The Discussion section is replete with information bullets, and speculative guesses about the data presented, but not really germane or tied to the data presented.  Unclear to this reviewer how the authors can cite "thrombosis risk" or any kinda of "physiological activity".  The conclusion was not supported by any evidence presented in the results.

In my opinion, the authors should decide what the focus of their study/manuscript should be, then stick to that script. 

Author Response

The authors describe their work using clot wave analysis in Hemophilia A patients and various therapies. I found this manuscript difficult to follow the intent and purpose.  There are bullets of information provide, but I have to try and connect those dots.  

Comment 1.

The claim that there is no mechanism for measuring inhibitors in patients on emicizumab is false, as chromogenic method using alternative factor has been described.

Response 1. Line 59: “using the APTT one-stage method” was added. Line 62: “Furthermore, the new chromogenic substrate assay is not affected by emicizumab and is suitable for use in determining FVIII coagulant activity and inhibitors [11].

Comment 2.

Unclear whether the factor VIII levels present in the table are upon diagnosis or measured during the study. If the former, that information is not germance.

Unclear when the patients were treated with the therapy in relationship to the drug level, as many FVIII levels were really low, so it appears that these are random collections, versus peak or trough collections.

Response 2. “FVIII activity” was changed to “Basal FVIII activity”. “Basal FVIII activity was measured in patients who had not been treated with FVIII concentrate,” was added to the Table legend. 

Comment 3.

Materials and methods are poorly organized.  Describe the sample collection first, the sample processing next.  Indicate when the samples were tested in relationship to collection time.

Response 3. The Materials and methods section has been revised.

Comment 4.

If surrogate or contrive samples were created, including calibration curves, describe how that was done.  Unclear what emicizumab concentrations were assessed in this study, and how they were assessed.  How much emicizumab was in the calibration curve (Figure 1)

Response 4. The text, “emicizumab (final concentration, 0.15 mg/ml)” has been added to the Results and Figure legend.

Comment 5.

Next describe the testing methods.  As in reading throughout this document, it appears on the peak height is germane to this manuscript.  As such, provide a figure and indicate with an arrow, which peak height is being used for this purpose.

Response 5. A figure with an arrow has been added.

Comment 6.

The statistical analysis is either not well described, or was not provided.  There are numerous indications where "significant" was used, yet no numeric or statistical information provided to support the use of that statistical term.

Response 6. “Significant” or “significantly” has been changed to “marked” or “markedly” in cases without statistical analyses.  

Comment 7.

Unclear what defines "almost similar" regarding calibration curve described in Figure 1. It is unclear how this was generated. Was the same calibrator material used?  IF so, why does the x-axis reflect differences in calibrator value?  If not, then why not?

Response 7. These sentences have been fully revised.

Comment 8.

Figure 2 appears to demonstrate equivalence between acceptable FVIII measurements (chromogenic) and the proposed platforms.  However, correlation, slope are more useful (and required measures by regulatory authorities) that this simple plot.

Unclear the purpose for Figure 3a or 3b.  

Unclear all the lines around the correlation line.

Response 8. Figure 3 (Figure 4 in revised version) showed the correlation and slops. Figure 3 has now been revised.

Comment 9.

As it appears these methods for ascertaining FVIII are different, what is the normal reference range for these platforms when using the same donors?

Response 9. Normal reference ranges have been added to the Materials and Methods section.

Comment 10.

Figure 4: which are of the CWA is used for FVIII analysis.  What is the purpose of this curve other to demonstrate different CWA between the two patients?  Why list all the other clot waves if they are not part of the analysis?   Easier to distinguish clot waves if more than different shades of blue and red are used. 

Response 10. An explanation of the purpose has now been added. The explanation text of “solid line, patient; dotted line, healthy volunteer” was already shown.

Comment 11.

Figure 5. The bottom curves need more explanation including where did this data come from as the navy (blue) line makes no sense as presented.  Is this a single HA patient with different FVIII over time after therapy?  Why is the overlay "a healthy volunteer" and not reflective of a normal range?  What is the purpose of this presentation?

Response 11. The figure has been revised. The dotted line shows the average line from 20 healthy volunteers. The purpose has been explained in the legend of the figure.

Comment 12.

The Discussion section is replete with information bullets, and speculative guesses about the data presented, but not really germane or tied to the data presented.  Unclear to this reviewer how the authors can cite "thrombosis risk" or any kinda of "physiological activity".  The conclusion was not supported by any evidence presented in the results.

Response 12. The Discussion section has been fully revised.

Comment 13.

In my opinion, the authors should decide what the focus of their study/manuscript should be, then stick to that script. 

Response 13. This manuscript has been fully revised.

Reviewer 3 Report

The authors are presenting data about different measurements of FVIII activity in patients with haemophilia A treated with different VIII concentrates and emicizumab. They are concluding that patients treated with FVIII concentrates have transient increased risk of thrombosis when they reach 100% FVIII activity, as they are affected by the thrombin burst.

Although this data may be interesting for the haemophilia community, revision of the manuscript is needed before publishing.

The clinical data and treatment details of patients are lacking. Without this data, the results cannot be interpreted and do not have clinical value. Type of prophylactic treatment, product and the dose at the time of blood sampling should be clearly stated for all laboratory tests described. It is important to emphasize if patients on emicizumab also received FVIII concentrate. Also, it should be noted for every experiment if emicizumab was added to the test, or plasma from patients on emicizumab was used, as it is not evident.

Appropriate tests for patients on emicizumab should be used. The difference between emicizumab concentration and FVIII activity in patients receiving emicizumab should be clearly distinguished.

Specific comments:

Several facts about haemophilia treatment are incorrect. Prophylactic treatment is standard for treatment of all patients with severe haemophilia, not only children. EHL products are already widely available. Any factor concentrate can elevate FVIII more that 100%.  "Haemostatic ability" is term not usually used to describe efficacy of haemostasis.

"Emicizumab.... is useful for life-threatening haemophilic patients with inhibitors for FVIII and it can reduce the injection frequency for FVIII products in the treatment of haemophilia A." This is not true description of emicizumab. Patients with inhibitors may have life-threatening bleedings, however, emicizumab is indicated in all patients with haemophilia A and inhibitors. Moreover, emicizumab is monoclonal antibody and not FVIII product. Therefore, it cannot be written that emicizumab reduces the frequency of FVIII products injection.

"FVIII inhibitors cannot be evaluated in patients treated with emicizumab." This statement is not true. Inhibitors can be evaluated, the bovine chromogenic Bethesda assay (CBA) should be used to measure FVIII activity and FVIII inhibitors.

What are exactly "hemophilia-related diseases"? This term should be modified.

"Regular replacement therapy" should be changed with the precise description of concentrate type. In some countries EHL are considered regular replacement therapy.

Figure 1. Was plasma from healthy donor used in this experiment?

Figure 2. Whose blood samples were used? No treatment, factor FVIII concentrates?

Figure 3.¸What does FVIII activity represents in patients on emicizumab?

Figure 5. Legend should also describe who are a), b) and c).

Author Response

The authors are presenting data about different measurements of FVIII activity in patients with haemophilia A treated with different VIII concentrates and emicizumab. They are concluding that patients treated with FVIII concentrates have transient increased risk of thrombosis when they reach 100% FVIII activity, as they are affected by the thrombin burst.

Although this data may be interesting for the haemophilia community, revision of the manuscript is needed before publishing.

Comment 1.

The clinical data and treatment details of patients are lacking. Without this data, the results cannot be interpreted and do not have clinical value. Type of prophylactic treatment, product and the dose at the time of blood sampling should be clearly stated for all laboratory tests described. It is important to emphasize if patients on emicizumab also received FVIII concentrate. Also, it should be noted for every experiment if emicizumab was added to the test, or plasma from patients on emicizumab was used, as it is not evident.

Response 1. The Materials and Methods section has been fully revised.

Comment 2.

Appropriate tests for patients on emicizumab should be used. The difference between emicizumab concentration and FVIII activity in patients receiving emicizumab should be clearly distinguished.

Response 2. The emicizumab concentration may not be useful in clinical practice. The FVIII activity in patients treated with emicizumab can be measured using a CWA-TT but not an APTT one-stage clotting assay. “FVIII activity” has been changed to “FVIII-like activity”. 

Specific comments:

Comment 3.

Several facts about haemophilia treatment are incorrect. Prophylactic treatment is standard for treatment of all patients with severe haemophilia, not only children. EHL products are already widely available. Any factor concentrate can elevate FVIII more that 100%.  "Haemostatic ability" is term not usually used to describe efficacy of haemostasis.

Response 3. The Introduction section has been revised.

Comment 4.

"Emicizumab.... is useful for life-threatening haemophilic patients with inhibitors for FVIII and it can reduce the injection frequency for FVIII products in the treatment of haemophilia A." This is not true description of emicizumab. Patients with inhibitors may have life-threatening bleedings, however, emicizumab is indicated in all patients with haemophilia A and inhibitors. Moreover, emicizumab is monoclonal antibody and not FVIII product. Therefore, it cannot be written that emicizumab reduces the frequency of FVIII products injection.

Response 4. These sentences have been revised.

Comment 5.

"FVIII inhibitors cannot be evaluated in patients treated with emicizumab." This statement is not true. Inhibitors can be evaluated, the bovine chromogenic Bethesda assay (CBA) should be used to measure FVIII activity and FVIII inhibitors.

Response 5. These sentences have been revised.

Comment 6.

What are exactly "hemophilia-related diseases"? This term should be modified.

Response 6. This sentence has been revised.

Comment 7.

"Regular replacement therapy" should be changed with the precise description of concentrate type. In some countries EHL are considered regular replacement therapy.

Response 7. The word “replacement” has been changed to “prophylaxis”.

Comment 8.

Figure 1. Was plasma from healthy donor used in this experiment?

Response 8. Yes the Materials and Methods and Results sections have now been revised. 

Comment 9.

Figure 2. Whose blood samples were used? No treatment, factor FVIII concentrates?

Response 9. Platelet-poor plasma and platelet-rich plasma were used from all patients, regardless of treatment. The Results and Figure Legends have been revised.

Comment 10.

Figure 3.¸What does FVIII activity represents in patients on emicizumab?

Response 10. “FVIII activity” has been changed to “FVIII-like activity”. The CWA-TT and chromogenic assay are not markedly affected by emicizumab treatment.

Comment 11.

Figure 5. Legend should also describe who are a), b) and c).

Response 11. Figure 5 (Figure 6 in revised version) has been revised.